# Type II DNA Topoisomerases Cause Spontaneous Double-Strand Breaks in Genomic DNA

**DOI:** 10.3390/genes10110868

**Published:** 2019-10-30

**Authors:** Suguru Morimoto, Masataka Tsuda, Heeyoun Bunch, Hiroyuki Sasanuma, Caroline Austin, Shunichi Takeda

**Affiliations:** 1Department of Radiation Genetics, Graduate School of Medicine, Kyoto University, Yoshida Konoe, Sakyo-ku, Kyoto 606-8501, Japanhiroysasa@rg.med.kyoto-u.ac.jp (H.S.); 2Department of Applied Biosciences, College of Agriculture and Life Sciences, Kyungpook National University, Daegu 41566, Korea; hbunch@knu.ac.kr; 3The Institute for Cell and Molecular Biosciences, the Faculty of Medical Sciences, Newcastle University, Newcastle upon Tyne NE2 4HH, UK; caroline.austin@newcastle.ac.uk

**Keywords:** topoisomerase II, genotoxicity, cell cycle, estrogen, breast cancer, transcription, BRCA1, BRCA2

## Abstract

Type II DNA topoisomerase enzymes (TOP2) catalyze topological changes by strand passage reactions. They involve passing one intact double stranded DNA duplex through a transient enzyme-bridged break in another (gated helix) followed by ligation of the break by TOP2. A TOP2 poison, etoposide blocks TOP2 catalysis at the ligation step of the enzyme-bridged break, increasing the number of stable TOP2 cleavage complexes (TOP2ccs). Remarkably, such pathological TOP2ccs are formed during the normal cell cycle as well as in postmitotic cells. Thus, this ‘abortive catalysis’ can be a major source of spontaneously arising DNA double-strand breaks (DSBs). TOP2-mediated DSBs are also formed upon stimulation with physiological concentrations of androgens and estrogens. The frequent occurrence of TOP2-mediated DSBs was previously not appreciated because they are efficiently repaired. This repair is performed in collaboration with BRCA1, BRCA2, MRE11 nuclease, and tyrosyl-DNA phosphodiesterase 2 (TDP2) with nonhomologous end joining (NHEJ) factors. This review first discusses spontaneously arising DSBs caused by the abortive catalysis of TOP2 and then summarizes proteins involved in repairing stalled TOP2ccs and discusses the genotoxicity of the sex hormones.

## 1. Introduction

Double-strand breaks (DSBs) are the most genotoxic type of DNA lesion, and a single unrepaired DSB can trigger apoptosis. If misrepaired, DSBs often cause deletions and chromosome translocation, which may trigger carcinogenesis [1]. Lethal DSBs are generated not only by environmental mutagens such as ionizing-radiation but also by endogenous cellular processes, with several DSBs generated per cell cycle as a consequence of replication blockage at spontaneously arising DNA lesions (Figure 1) [2,3]. They include damaged nucleotides on template strands and single-strand breaks. Homology-directed repair (HDR) repairs DSBs using the homologous sister chromatid as the template DNA for repair. (Figure 1) [4]. Breast cancer susceptibility gene 1 (BRCA1) and breast cancer susceptibility gene 2 (BRCA2), are tumor suppressor proteins that play a key role in HDR (Figure 1) in cycling cells, and the loss of these proteins causes accumulation of DSBs. Mutations in the *BRCA1* and *BRCA2* genes predispose carriers to a higher risk of breast and ovarian cancer [5,6]. One major question in BRCA biology is why carcinogenesis occurs mainly in estrogen-regulated organs, arising in the epithelial cells in the mammary and ovarian ducts.

## 2. The Consequences of Failure of Catalysis by DNA Topoisomerase I and II

DNA replication and transcription overwind duplex DNA ahead of replication forks and transcription bubbles, respectively, and the resulting torsional stress would eventually stop DNA and RNA polymerases. Topoisomerase I (TOP1) cleaves one strand of a DNA double helix, allowing one end of the resulting single-strand break (SSB) to rotate around the uncut strand prior to re-ligation of the SSB and thereby resolving torsional stress [7]. TOP1 fails to re-ligate SSBs when TOP1 cleaves at ribonucleotides embedded in DNA and near ultraviolet (UV) lesions, abasic sites, or SSBs [7,8,9,10]. This failure causes the formation of stabilized TOP1 cleavage complexes (TOP1ccs), where SSBs are covalently attached to TOP1 at their 3′ end (Figure 2A). Tyrosyl-DNA phosphodiesterase 1 (TDP1) as well as tyrosyl-DNA phosphodiesterase 2 (TDP2) accurately removes such trapped TOP1 adducts from SSBs [11,12,13]. Loss of these proteins causes spinocerebellar ataxia with axonal neuropathy [14,15], which is associated with the endogenous accumulation of stabilized TOP1ccs in neural tissue [16,17,18], indicating that TOP1 is frequently trapped in the genomic DNA.

Similar to TOP1, topoisomerase II (TOP2) can be trapped on genomic DNA, which leads to the formation of stabilized TOP2 cleavage complexes (TOP2ccs) (Figure 2B). There are two human TOP2 paralogs, topoisomerase IIA (TOP2A) and topoisomerase IIB (TOP2B) [19]. TOP2A is expressed only in cycling cells while TOP2B is ubiquitously expressed. TOP2A is essential for DNA replication, chromosome condensation, and separation of sister chromatids. Both TOP2A and TOP2B are involved in transcriptional initiation and elongation [20]. The two TOP2 enzymes resolve DNA catenanes by catalyzing the transient formation of DSBs, which is followed by enzymatic religation of the broken strands (Figure 2B). Thus, TOP2 has the capacity to rejoin cleaved DNA ends through intrinsic intramolecular ligation activity [21]. Transient DSB formation allows an intact DNA duplex to pass through the DSB. During such transient DSB formation, TOP2 becomes covalently bound to the 5′ DNA end of the break, forming TOP2-DNA cleavage-complex intermediates (TOP2ccs). An anti-cancer TOP2 poison, etoposide, stabilizes TOP2ccs, and this can trigger cell death (Figure 2B) [22] and may cause secondary leukemia by inducing chromosome translocations [23]. Exposure to etoposide at an elevated temperature, 39 °C, synergistically suppresses cellular viability [24]. Likewise, the exposure to etoposide together with an inhibitor against heat shock protein 90 (HSP90), a protein-folding chaperone, which transiently binds other proteins to protect them from misfolding and aggregation, also synergistically suppresses cellular viability. These data suggest that the conformation of TOP2 is sensitive to a temperature shift and its maintenance requires HSP90. Nonetheless, it was previously believed that the abortive catalysis of TOP2 occurs extremely rarely under physiological conditions.

Spontaneously arising stabilized TOP2ccs have not been considered as a serious threat to genome instability for the following reasons. While TDP2 accurately removes 5′ TOP2 adducts, a defect in TDP2 does not cause prominent carcinogenesis or genome instability in patients or mice [25,26,27]. However, subsequent studies show that conditional inactivation of MRE11 nuclease activity causes the endogenous accumulation of stabilized TOP2ccs in human cells leading to prominent genome instability [28]. This study highlights the abortive catalysis of TOP2 in addition to replication blockage on a damaged template strand as the two major mechanisms for spontaneously arising DSBs (Figure 2C). This review will discuss the accumulation of such stalled TOP2ccs and the crucial role that MRE11 plays in repair of DSBs.

## 3. Removal of 5′ TOP2 Adducts to Restore Ligatable DNA Ends is Necessary for DSB Repair by NHEJ

There are two major DSB repair pathways, HDR and nonhomologous end joining (NHEJ). HDR functions in the S and G2 phases, and accurately repairs DSBs using sister chromatids to provide an intact DNA template. NHEJ, on the other hand, operates in all cell cycle phases. The choice between the major DSB repair pathways, HDR versus NHEJ, can be mediated by resection, to produce 3′ single-strand (ss) tails at DSBs (Figure 3A). This resection promotes HDR and suppresses NHEJ. DSB resection is initiated by endonucleolytic cleavage by MRE11, which is physically associated with RAD50 and NBS1 (MRN complex), followed by extensive resection by two exonucleases, DNA2 and EXO1 [4]. BRCA1 and CtIP significantly contribute to DSB resection by interacting with MRE11 at DSB sites. RAD51 recombinase polymerizes on 3′ ss tails, and this polymerization is promoted by breast cancer susceptibility genes 1 and 2 (BRCA1 and BRCA2). Approximately 80% of ionizing-radiation-induced DSBs are repaired by NHEJ even in the G2 phase [29,30], indicating the dominant role played by NHEJ in DSB repair when DSBs are not caused by stalled DNA replication (Figure 2C, left).

HDR efficiently removes 5′ TOP2 adducts at its initial step, DSB resection by MRE11 endonuclease (Figure 3B). In contrast with HDR, NHEJ requires 3′-OH and 5′-phosphate DSB ends, termed ‘clean’ DSB ends, for direct ligation [31]. NHEJ is initiated by binding of KU70/KU80 proteins and is completed by ligation by DNA ligase 4 (LIG4) (Figure 3C) [32]. Ligation by LIG4 requires ligatable DSB ends [31,33], including DSBs carrying ribonucleotides at their ends [34], and NHEJ requires prior removal of blocking adducts and restoration of ligatable DSB ends. TDP2 and MRE11 remove 5′ TOP2 adducts prior to direct ligation by NHEJ (Figure 3C). The removal by TDP2 has two routes either digestion of 5′ TOP2 adducts by the proteasome prior to TDP2 action or interaction of the TOP2ccs with the sumo ligase ZATT (ZNF451) to facilitate TDP2 action [35,36]. The contribution of MRE11 to the removal of 5′ TOP2 adducts is considerably larger than that of TDP2, as evidenced by the fact that the loss of MRE11, but not the loss of TDP2, results in endogenous accumulation of TOP2ccs [28].

## 4. MRE11 Contributes to DSB Repair through Two Distinct Mechanisms: The Removal of Stalled TOP2ccs and DSB Resection in HDR

HDR is essential for repairing lethal DSBs occurring during DNA replication. Conditional inactivation of RAD51 recombinase causes spontaneously arising lethal DSBs in every cell and immediately kills cycling cells [37]. The loss of MRE11, RAD50, or NBS1 causes a dramatic increase in the number of spontaneously arising DSBs leading to cell death [37]. Conditionally generated MRE11-null deficient mice (*MRE11*^−/−^) and nuclease-deficient *MRE11*^−/*H129N*^ mice display a very similar phenotype: hypersensitivity to ionizing radiation, very severe genome instability, and cellular senescence, resulting in cellular mortality [38]. These data have been interpreted as evidence that a defect in DSB resection by the MRE11 nuclease activity in HDR is solely responsible for the genome instability seen in the MRE11-deficient mice. However, it remains elusive whether the defective DSB resection is responsible for the severe phenotype of the *MRE11*^−/*H129N*^ and *MRE11*^−/−^ mice for the following reasons. These mice showed only a ~50% decrease in X-ray induced RAD51-focus formation compared with wild-type controls [38]. It is unlikely that the ~50% decrease causes the genome instability seen in the MRE11 mutant mice because only a two-fold reduction of DSB resection does not affect the overall efficiency of HDR including heteroallelic HDR [39]. Taken together, mammalian cells perform excessive DSB resection, and it remains unclear whether a modest defect in DSB resection causes the very severe genome instability in MRE11-deficient cells.

Several lines of evidence have suggested that MRE11 contributes to genome stability through a mechanism other than DSB resection. Although the conditional inactivation of MRE11 does not impair DSB resection in cell lines such as human TK6 and chicken DT40 B cell lines, the MRE11-deficient cells show spontaneously arising chromosomal breaks in mitotic chromosome spreads leading to cell death [37,39]. In Saccharomyces cerevisiae, MRE11 is dispensable for HDR-dependent repair of ‘clean’ DSBs carrying 5′-phosphate and 3′-hydroxyl termini [40]. However, MRE11 significantly contributes to cellular tolerance of ionizing-radiation [4,41], which induces ‘dirty’ DSBs associated with non-physiological chemical adducts. The data suggest that the MRE11 nuclease activity removes chemical adducts from ‘dirty’ DSBs and restores ‘clean’ DSBs. This idea is supported by the following reverse genetic studies. The nuclease activity of *Schizosaccharomyces pombe* MRE11 is involved in the removal of TOP1 and TOP2 adducts from 3′ and 5′ ends, respectively, in vivo [42]. Likewise, human MRE11 nuclease activity, as well as NBS1, is required for the removal of 5′ topoisomerase adducts from DSBs in vivo [28,43]. Biochemical studies show that non-covalent DNA-bound KU70/80 proteins stimulate the endonuclease activity of MRE11 cleaving up to 30 nucleotides inside from DSB ends, leading to removal of blocking adducts from both 3′ and 5′ termini of DSBs [44,45] (reviewed in [46]) (Figure 3C). In summary, the endonuclease activity of MRE11 may have two distinct roles, DSB resection for HDR and removal of non-physiological chemical adducts from DSB ends.

The conditional inactivation of MRE11 nuclease causes a number of spontaneously arising chromosome breaks in mitotic chromosome spreads and kills all cycling cells [28,37,47]. It was believed that this lethality results solely from a defect in HDR, which rejoins multiple lethal DSBs (Figure 1) in each cell cycle [4]. However, the conditional inactivation of MRE11 nuclease activity also causes the endogenous accumulation of stalled TOP2ccs in human cells in addition to cellular mortality [28]. Moreover, overexpression of TDP2 suppresses by ~25% both the lethality and prominent genome instability during the depletion of MRE11-nuclease activity [28]. Thus, the lethality of MRE11-deficient cells results not only from defective HDR but also from the endogenous accumulation of stabilized TOP2ccs. These data indicate that stabilized TOP2ccs have the potential to lead to the generation of lethal DSBs.

Neuron-specific loss of NBS1 also causes endogenous DSB accumulation in the adult brain of mice [28], indicating that the MRE11 complex plays a critical role in the elimination of 5′ TOP2 adducts even in postmitotic cells. The viability of the MRN-deficient post-mitotic cells indicates that the number of spontaneously arising TOP2ccs is not large enough to cause mortality in non-cycling cells but might impair neural function [47,48]. In summary, MRE11 plays dominant roles in repairing the two major sources of spontaneously arising DSBs, abortive catalysis of TOP2 and replication collapse at damaged template strands (Figure 2C).

## 5. Abortive TOP2B Catalysis Occurs during Early Transcriptional Responses to Estrogens

While TOP2A is essential for cell division, homozygous *TOP2B*^−/−^ murine embryos are born displaying no apparent morphological abnormality [49], indicating that TOP2B is dispensable for cellular proliferation. Detailed examination of *TOP2B*^−/−^ embryos and brain-specific TOP2B-deficient mice revealed neurological defects, such as defects in the innervation of both motor neurons to skeletal muscles and sensory neurons to the spinal cord [49,50]. *TOP2B*^−/−^ new-born mice die due to neurological defects resulting in lack of innervation to the diaphragm. These data indicate that TOP2B activity is required for proper development of neural networks.

A number of studies have suggested that TOP2B controls transcriptional initiation as well as elongation. Analysis of TOP2B ChIP-seq profiles indicates enrichment of TOP2B binding within the promoters and gene bodies of transcribed genes [20,51,52,53,54,55,56,57,58,59,60,61]. TOP2B contributes to neuronal development by facilitating the transcription elongation of the neuronal genes that are longer than 200 kb [51]. In addition to transcriptional elongation, TOP2B has been reported to be involved in transcriptional initiation and RNA polymerase II (Pol II) promoter-pause release upon physiological stimulations including androgens, estrogens, insulin, glucocorticoids, N-methyl-d-aspartate (NMDA), retinoic acid, heat shock, and serum [55,59,62,63,64,65,66]. Such signal-dependent transcriptional activation mediated by the mechanism called Pol II promoter-proximal pausing (Figure 4). This mechanism ensures the prompt and robust transcription of early response genes and some non-coding RNA genes immediately after stimulation with ligands and stresses [67,68,69]. In transcriptional activation of early response genes, TOP2B causes DSB formation, which is evidenced by the recruitment of NHEJ factors to transcriptional initiation sites of stimulus-inducible genes (Figure 4) [59,62,63,66,70,71,72,73].

Pol II promoter pausing involves multiple layers of regulations exerted by protein and nucleic acid factors that control Pol II enzyme processivity and nucleosome architecture [68,74,75]. In particular, recent studies have suggested that Pol II pausing is influenced by DNA torsion and that either negative or positive supercoiling can prevent Pol II from translocating on the DNA (Figure 4, Step 1) [76,77,78,79]. Biochemical analyses have demonstrated that TOP2B functions to resolve positive supercoiling ahead of transcribing Pol II [7,80,81]. Contributed to by the +1 nucleosome, a characteristic of the signal-dependent genes utilizing Pol II pausing [82,83], the positive supercoiling is likely to be generated in the early elongation step in these genes (Figure 4, Step 2) [78,81]. It is thus speculated that the catalytic activity of TOP2B to remove the supercoiling is required for Pol II pause release and productive transcription [7,59,66,74,84]. An interesting aspect of TOP2B function in Pol II pause release is that a DSB and DNA damage response involving diverse DNA repair factors are provoked in the process (Figure 4, Step 3), whereas TOP2B typically reseals the DSB immediately after resolving DNA torsion, without needing DNA repair factors. Therefore, it has been hypothesized that the TOP2B-mediated DSB during transcriptional pausing release is rather a persistent one that leads to the induction of the DNA damage response [7].

It is not yet clear whether or not the DSB introduced by TOP2B to relieve supercoiling plays additional physiological roles. An unsolved question is whether the TOP2B-mediated DSBs that require NHEJ for their rejoining have a physiological role in transcription regulation. The DSB formation might be programmed and contribute to robust transcriptional control (reviewed in [20,73]). Another possibility is that TOP2B-mediated DSBs are generated by non-physiological abortive catalysis of TOP2B, which is a stochastic event [85]. This idea is supported by the following data. There are two well-known examples of ‘programmed DSBs’. One triggers V(D)J recombination in the immunoglobulin variable gene [86], and the other is induced by SPO11, a TOP2-like enzyme, to initiate meiotic homologous DNA recombination (HR) [87,88]. Both impaired formation of typical programmed DSBs and defects in subsequent DSB repair cause the same phenotype. For example, the absence of the RAG1/RAG2 DNA cleavage enzyme [89] and a defect NHEJ-mediated DSB repair cause the same phenotype, impaired lymphocyte development. By contrast, the lack of innervation to the diaphragm in new-born *TOP2B*^−/−^ mice [49,50] is not seen in mice deficient in KU70, a protein essential for NHEJ [90,91]. In other words, the loss of TOP2B-dependent programmed DSBs causes the phenotype very different from that of the mice deficient in subsequent DSB repair. Thus, it remains to be proven whether or not the DSB introduced by TOP2B plays additional physiological roles other than relieving supercoiling. Nonetheless, the following data support the physiological roles. The inhibition of NHEJ by a DNA-PKcs inhibitor causes a deregulated Pol II pause release [62,63,64,65,66]. We also found that the loss of DNA-PKcs caused an increase in the transcription of *c-MYC* gene at 30 min after addition of estrogens (manuscript in preparation). These data indicate that TOP2B-mediated stable DSB formation and its subsequent rejoining by NHEJ may have a physiological role in immediate transcriptional responses. In summary, the recruitment of NHEJ factors to TOP2B mediated catalytic sites in early transcriptional responses (reviewed in [73]) supports two alternative scenarios: the physiological role played by NHEJ in rejoining programmed DSBs and, the very frequent occurrence of abortive catalysis of TOP2B that affects immediate transcriptional response to ligands.

## 6. BRCA1 and BRCA2 Promote MRE11-Mediated Removal of 5′ TOP2 Adducts from Pathological TOP2ccs

Recent studies revealed that BRCA1 and BRCA2 promote the repair of DSBs caused by abortive catalysis of TOP2. Before describing this new finding, this paragraph and the next one summarize previous key findings. Monoallelic germline mutations in the *BRCA1* and *BRCA2* genes predispose carriers to a high incidence of breast and ovarian cancer [92]. Loss of heterozygosity (LOH) causing biallelic BRCA gene inactivation provokes genomic instability and greatly enhances carcinogenesis (reviewed in [6]). Both BRCA1 and BRCA2 play a critical role in HDR of DSBs [93,94] as well as in protecting stalled replication forks from the fork collapse (Figure 1) [95,96,97,98]. Defects in the functionality of BRCA1 or BRCA2 increase the rate of mutagenesis, but no mutations in any specific genes account for oncogenesis selectively in estrogen-regulated tissues [99,100]. Considering that the occurrence of LOH events is extremely rare, inactivation of BRCA1 and BRCA2 by LOH may dramatically enhance the carcinogenesis of the estrogen-regulated tissues. Since HDR (Figure 1) plays an essential function in all cycling cells, it remains unclear why the carcinogenesis caused by defects in BRCA1 and BRCA2 manifests in such a highly tissue-restricted manner.

In addition to HDR (Figure 5A), BRCA1 and BRCA2 contribute to the maintenance of chromosomal DNA by suppressing the accumulation of RNA: DNA hybrid structures, which are known as R-loops (Figure 5B) and by promoting the repair of stalled TOP2ccs (Figure 5C). R-loop occurs co-transcriptionally at GC-rich sequences including G-quadruplex-containing sequences. R-loop formation is facilitated by SSBs and DSBs (reviewed in [101]) and seems to be induced by Pol II promoter pausing [102]. Replication blockage by R-loops causes genome instability and abnormal recombination, and it is believed that R-loops are not a serious threat to genome instability in G0/G1 phases (reviewed in [103]). Accumulating evidence implicate BRCA1 and BRCA2 in the turnover of R-loops (reviewed in [6]). BRCA1 removes R-loops by associating with Senataxin, a helicase implicated in the resolution of R-loops [104,105]. BRCA2 may bind to the branched structure formed at RNA: DNA hybrids and facilitate the access of Ribonuclease H1, an enzyme that selectively digests the RNA portion of an R-loops [106,107]. A recent study showed that BRCA2 binds to Pol II and controls the release of Pol II from promoter-proximal pausing sites [108]. BRCA2 inactivation results in accumulation of R-loops and DNA breakage at these sites. BRCA1, which was previously shown to co-immunoprecipitate with Pol II [109], also suppresses the accumulation of R-loops specifically at promoter-proximal pausing sites [109]. Interestingly, this accumulation is detectable only in luminal epithelial cells of murine mammary glands [110]. These data suggest BRCA1 and BRCA2 prevent the oncogenesis of mammary epithelial cells by inhibiting abnormal R-loop formation (Figure 5B).

The vast majority of mammary epithelial cells are in the G0/G1 phases, and the following data suggest that BRCA1 and BRCA2 may repair pathological TOP2ccs in these phases (Figure 5C). A defect in BRCA1 in mice causes embryonic lethality while mice deficient in both BRCA1 and 53BP1 are viable showing a rescue of the HDR defect in BRCA1 mutant cells [111]. These viable mice still constitutively manifest prominent genomic instability [111]. Moreover, despite the fact that HDR is suppressed in the G1 phase BRCA1 still accumulates at etoposide-induced DNA damage sites [112]. These data indicate that BRCA1 contributes to genome stability through a mechanism other than HDR. This idea is supported by data that the loss of BRCA1 is associated with an increase in the number of etoposide-induced TOP2ccs in the G1 phase [112] (Figure 3). Thus, BRCA1 promotes the repair of etoposide-induced DSBs independent of HR [112,113]. Etoposide-dependent physical interaction between BRCA1 and MRE11 in G1-arrested cells and cycling ones [114,115,116] indicates that BRCA1 stimulates the endonuclease activity of MRE11 and promotes the removal of 5′ TOP2 adducts preceding NHEJ (Figure 3C). BRCA2 also promotes the removal of 5′ TOP2 adducts (manuscript in preparation). In summary, BRCA1 and BRCA2 contribute to genome maintenance through at least three mechanisms. First, the two proteins promote HDR for rejoining DSBs occurring during DNA replication and also for protecting stalled replication forks (Figure 1 and Figure 5A). Second, BRCA1 and BRCA2 promote the resolution of R-loops and inhibit their collision with DNA replication forks (Figure 5B). Finally, these proteins promote the rejoining of stalled TOP2ccs, DSBs resulting from abortive TOP2 catalysis, during the whole cell cycle (Figure 3C and Figure 5C).

## 7. Estrogens Cause Accumulation of Stalled TOP2ccs in BRCA1-Deficient Mammary Epithelial Cells

Estrogens are very potent growth factors and are involved in the proliferation of mammary glands and ovarian tissue. Estrogens stimulate cell proliferation by activating estrogen receptor alpha and beta (ERα and ERβ). Activated ERs serve as ligand-activated transcription factors and bind to estrogen-response elements (EREs) in ER target genes (Figure 6, Step 1). TOP2A and TOP2B are involved in the initiation of ligand-activated transcription [25,59,60,64,70,71,117,118,119] (reviewed in [7,20]). TOP2s have been shown to play a role in transcriptional control by steroid hormones including both androgen and estrogen hormones (Figure 6, Step 2 and 3) presumably by promoting interactions between transcriptional promoters and enhancer segments. Previous studies using chemical inhibitors and reverse genetic approaches, however, have not yet identified the exact function of TOP2s in ligand-activated transcription. This is due to functional overlap between the two TOP2s, functioning of TOP2s in both transcriptional elongation and initiation, and an essential role for TOP2A in cellular proliferation.

Exposure of the MCF-7 human mammary adenocarcinoma cell line to 1 nM estrogens induces DSBs, and this DSB formation depends on ERs [118,119]. Activated ERs stimulate the transcription of ER target genes leading to formation of R-loops (Figure 5B), which cause DNA cleavage by interfering with the progression of DNA replication forks during S phase [118,119]. BRCA1 prevents the accumulation of estrogen-induced DSBs by suppressing transcriptional DNA damage occurring at R-loops [104].

A recent study has indicated that activated ERs induce DSBs also in G1 phase [112]. This induction depends on TOP2B and does not require ongoing transcription or formation of R-loops. Remarkably, unrepaired DSBs are detectable even one day after a two-hour pulse exposure to estrogens in the absence of either BRCA1, BRCA2, or NHEJ [112]. The exposure to estrogen causes accumulation of unrepaired DSBs also in mammary epithelial cells of mice deficient in BRCA1 or NHEJ [112]. These results suggest the following mechanism for estrogen-induced DSBs and their repair. Activated ERs often cause the formation of stalled TOP2ccs in G0/G1 phases presumably at promoter and enhancer sequences (Figure 6, Step 4). Stalled TOP2ccs are quickly repaired by collaboration of BRCA1, BRCA2, Mre11, TDP2, and NHEJ (Figure 6, Step 5 and 6). In summary, there are two mechanisms for the induction of DSBs by estrogens: the induction of R-loops and their collision with DNA replication forks during the S phase (Figure 5B), and the induction of TOP2ccs throughout the cell cycle (Figure 5C). The above genotoxicity of estrogen is relevant to other ligands that cause immediate early transcriptional responses including physiological and medically relevant concentrations of glucocorticoid and dexamethasone, drugs widely prescribed to treat allergy and inflammation (manuscript in preparation). The genotoxicity of glucocorticoid and dexamethasone has not been appreciated because of efficient repair of the DSB caused by these chemicals in DNA-repair-proficient wild-type cells.

Immediate early transcriptional responses by a wide variety of ligands often cause the formation of stalled TOP2Bccs [59], and BRCA1 and BRCA2 promote the repair of these DSBs. Thus, inactivation of BRCA1 and BRCA2 may increase the genotoxicity of these ligands in a variety of tissues. A prominent question has been why does the inactivation of BRCA1 and BRCA2 selectively enhance the oncogenesis of the estrogen-regulated organs. One scenario is that estrogens strongly stimulate the proliferation of epithelial cells in mammary glands and the ovary by controlling estrogen-receptor-target genes, and promote the oncogenesis of these tissues (Figure 7). Stalled TOP2ccs not only induce DNA mutations but also might enhance transcriptional responses to estrogens by further activating the transcription of oncogenes or decreasing the expression of tumor suppressors (Figure 7). The idea that unrepaired DSBs enhance transcription of neighboring genes challenges the view that unrepaired DSBs suppress transcription from the promoter localized within several kb from the DSBs in an ataxia telangiectasia mutated (ATM) kinase-dependent manner [120,121]. However, unrepaired DSBs localized far beyond several kb from the transcriptional promoter can upregulate the immediate transcriptional response to ligands as evidenced by the data that a CRISPR/Cas9-mediated DSB at a TOP2B catalysis site actually upregulates transcriptional responses to a neurotransmitter [59]. A possible scenario is that estrogens cause abortive catalyzes of oncogenes perhaps in their transcriptional enhancer segments (Figure 7 Red arrows) and that the resulting stable DSBs might facilitate physical interactions between promoter and enhancer sequences. This scenario may help to understand molecular mechanisms for oncogenesis selectively in estrogen-regulated tissues in the absence of BRCA1 or BRCA2.

## 8. Conclusions and Unsolved Questions

Recent studies have demonstrated that abortive catalysis of TOP2s poses a major threat to genome stability in addition to DSB formation caused by replication blockage (Figure 2C). While DSBs caused by replication blockage are accurately rejoined exclusively by HDR, pathological TOP2ccs are repaired frequently by NHEJ, with rejoining often being associated with insertion and deletion mutations forming at joint/junction sequences. The endonucleolytic cleavage by MRE11 to remove 5′ TOP2 adducts generates 3′ overhang at DSB ends. If the ends are incompatible, they are processed until DSB ends have a ligatable configuration such as a few nucleotides microhomology, and microhomology-mediated NHEJ would lead to deletion mutations at joint/junction sequences. Considering a dominant role for MRE11 in the repair of stalled TOP2ccs [28], cells might have evolved the mechanism by which collaboration between MRE11 and NHEJ performs accurate rejoining without causing deletion mutations. It remains elusive by which mechanism NHEJ ensures accurate rejoining in transcription-coupled abortive TOP2B catalysis. The role of TOP2s in the regulation of transcription is the major unsolved question. Previous studies examined the effect of etoposide on the transcription of genes. However, this method does not clearly distinguish the role of TOP2 in transcriptional initiation from that in transcriptional elongation and TOP2-mediated DSB sites. In addition, the sequential order of events including the reception of activating signal and TOP2B activity during Pol II pause release is less defined in many signal-inducible genes. To address these questions, we propose that the effect of TOP2 depletion on ligand-mediated transcriptional activation in cells synchronized at G0/G1 phase needs to be studied. In addition, the mechanisms by which BRCA1 and BRCA2 facilitate the removal of TOP2ccs needs to be addressed in future studies.

## Figures and Tables

**Figure 1 genes-10-00868-f001:**
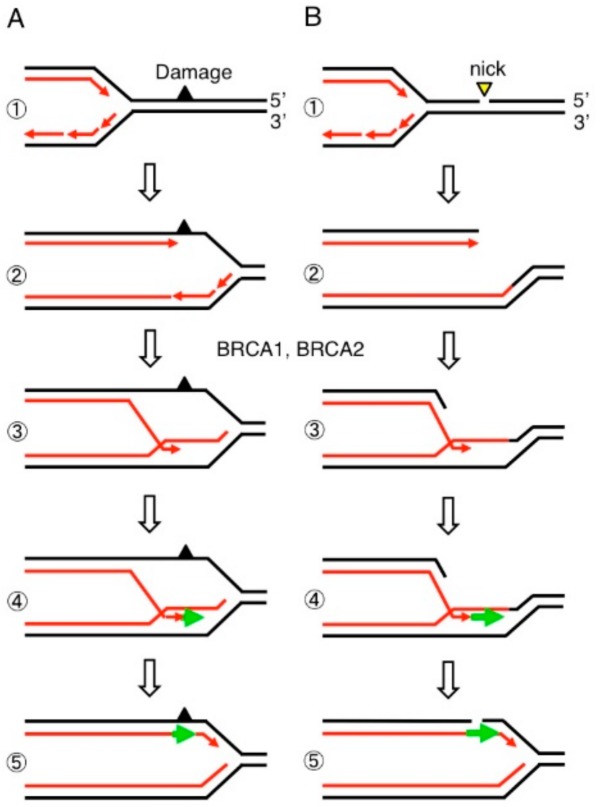
Release of replication blockage by breast cancer susceptibility gene 1 (BRCA1) and breast cancer susceptibility gene 2 (BRCA2). DNA replication forks often stall at both damaged nucleotides (**A**) and single-strand breaks (**B**) on template strands (Step 1). Homology-directed repair (HDR) proteins including BRCA1 and BRCA2 initiate strand invasion of the 3′ end of nascent DNA strands into the intact sister chromatid (Step 3) followed by DNA synthesis (Step 4, green arrow). This pathway allows the fork to move past the damaged nucleotides (**A**) and single-strand breaks (**B**) (Step 5).

**Figure 2 genes-10-00868-f002:**
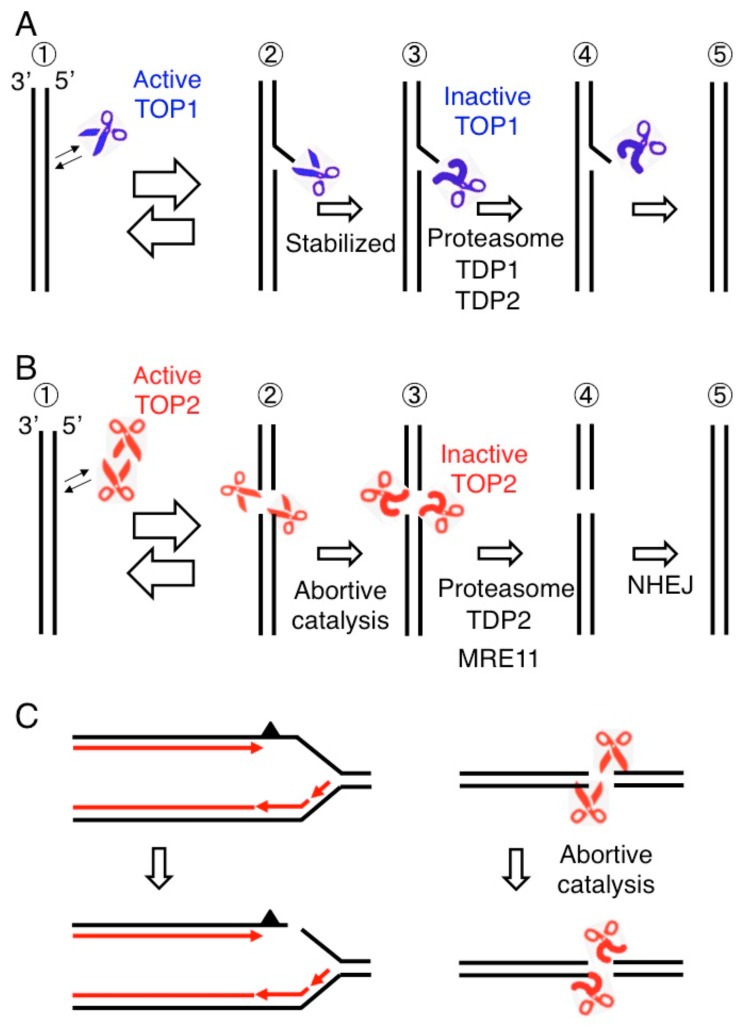
Abortive catalysis of topoisomerases causes DNA cleavage. (**A**) TOP1 cleaves one strand of a DNA double helix and religates the nick (Step 1 and 2). Topoisomerase I (TOP1) is covalently associated with 3′ end generating a TOP1 cleavage complex (TOP1cc shown in Step 2). TOP1 occasionally fails to religate, generating stable TOP1ccs (Step 3). 3′ TOP1 adducts are removed by the proteasome and TDP enzymes (Step 4) followed by religation of single-strand breaks (step 5). (**B**) A topoisomerase II (TOP2) homodimer generates a gated double-strand breaks (DSB) called a TOP2 cleavage complex (TOP2cc), where TOP2 covalently associates with 5′ end of DSB (Step 2). When TOP2 fails to religate TOP2ccs called abortive catalysis of TOP2 (Step 3), religation of the DSB is carried out by the removal of 5′ TOP2 adducts (Step 4) preceding nonhomologous end joining (NHEJ) (Step 5). (**C**) Two major sources of spontaneously arising DSBs, the blockage of replication forks leading to replication fork collapse (left) and abortive catalysis of TOP2 leading to the generation of stalled TOP2ccs (right).

**Figure 3 genes-10-00868-f003:**
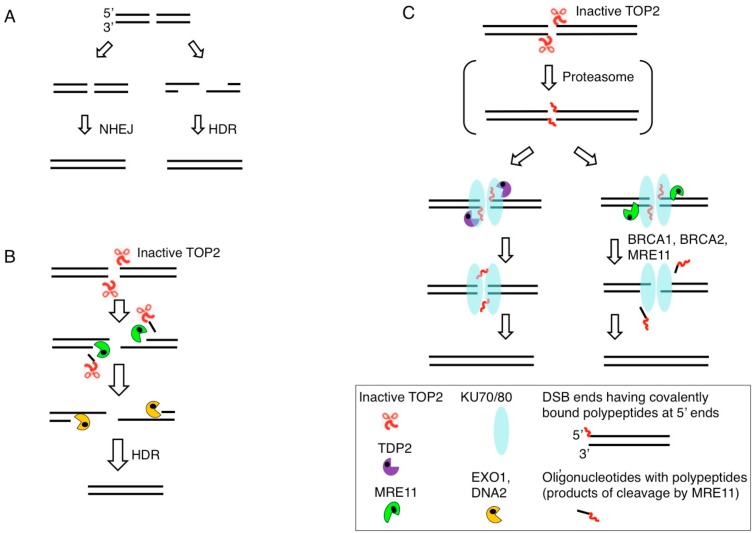
Removal of 5′ TOP2 adducts. (**A**) DSBs are repaired by HDR or NHEJ. DSBs occurring during DNA replication are repaired exclusively by HDR (Figure 1). The choice between HDR and NHEJ is determined by DSB resection, the generation of 3′ single-strand tails, which inhibit NHEJ and stimulate HDR. (**B**) DSB resection is carried out by the coordinated action of the MRE11, EXO1, and DNA2 nucleases. Endonucleolytic cleavage of 5′ strands by MRE11 near DSB ends removals 5′ TOP2 adducts together with oligonucleotides. (**C**) NHEJ ligates ‘clean’ DSBs having 5′-phosphate and 3′-hydroxyl moieties but not DSBs bearing blocking adducts. 5′ TOP2 adducts are partially digested by the proteasome generating DSBs carrying 5′ oligopeptides. 5′ oligopeptides are removed by tyrosyl-DNA phosphodiesterase 2 (TDP2) (Left) or MRE11 (Right). Resulting ‘clean’ DSBs are repaired by NHEJ, which is initiated by binding of the KU70/KU80 heterodimer to DSB ends.

**Figure 4 genes-10-00868-f004:**
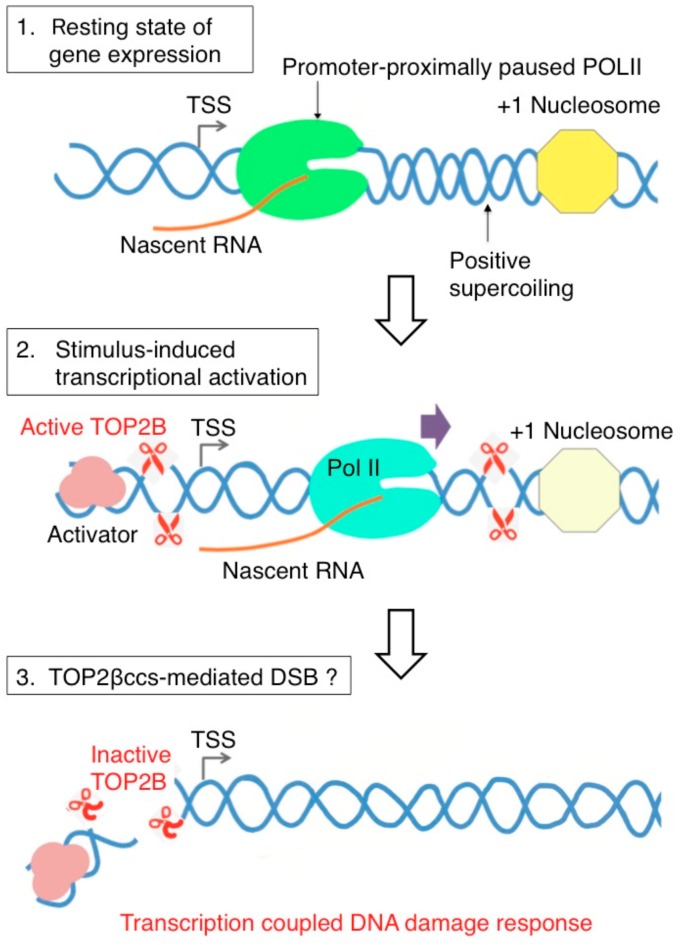
Model of topoisomerase IIB (TOP2B)-mediated DSB and DNA damage response signaling during RNA polymerase II (Pol II) pause release in stimulus-inducible protein-coding genes in humans. Top in the uninduced state of transcription, Pol II is paused between +25 and +100 from the transcription start site. The pausing is attributed to different elements including pausing-stabilizing transcription factors, the +1 nucleosome, and DNA structure and torsion. Positive supercoiling ahead of Pol II may require the function of TOP2B. Middle, transcription activation induced by various stimuli activates TOP2B to resolve DNA torsion in the promoter and gene body. Bottom, in this process, DSB could be formed from abortive catalysis of TOP2B, which occurs frequently in some genes. This may be responsible for DNA damage response signaling that has been observed in a number of stimulus-inducible genes in humans [59,66,70].

**Figure 5 genes-10-00868-f005:**
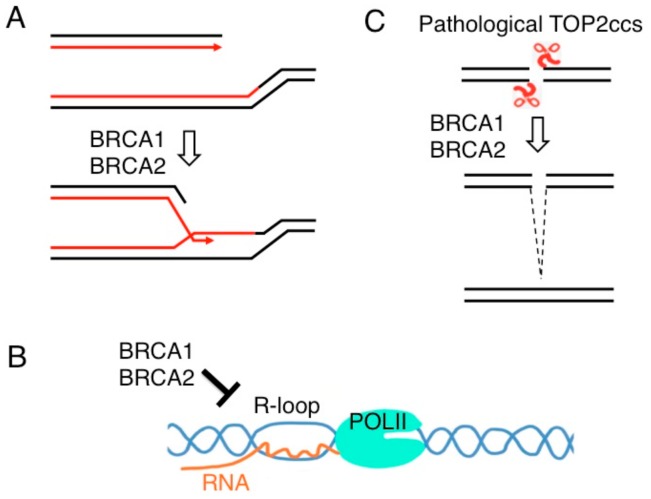
The roles of BRCA1 and BRCA2 in genome maintenance. (**A**) They play a critical role in the repair of DSBs formed during DNA replication by promoting strand exchange between two sister chromatids as shown in Figure 1. (**B**) BRCA1 and BRCA2 reduce the accumulation of transcription-associated R loop. (**C**) BRCA1 and BRCA2 promote the repair of stalled TOP2ccs as shown in Figure 3C.

**Figure 6 genes-10-00868-f006:**
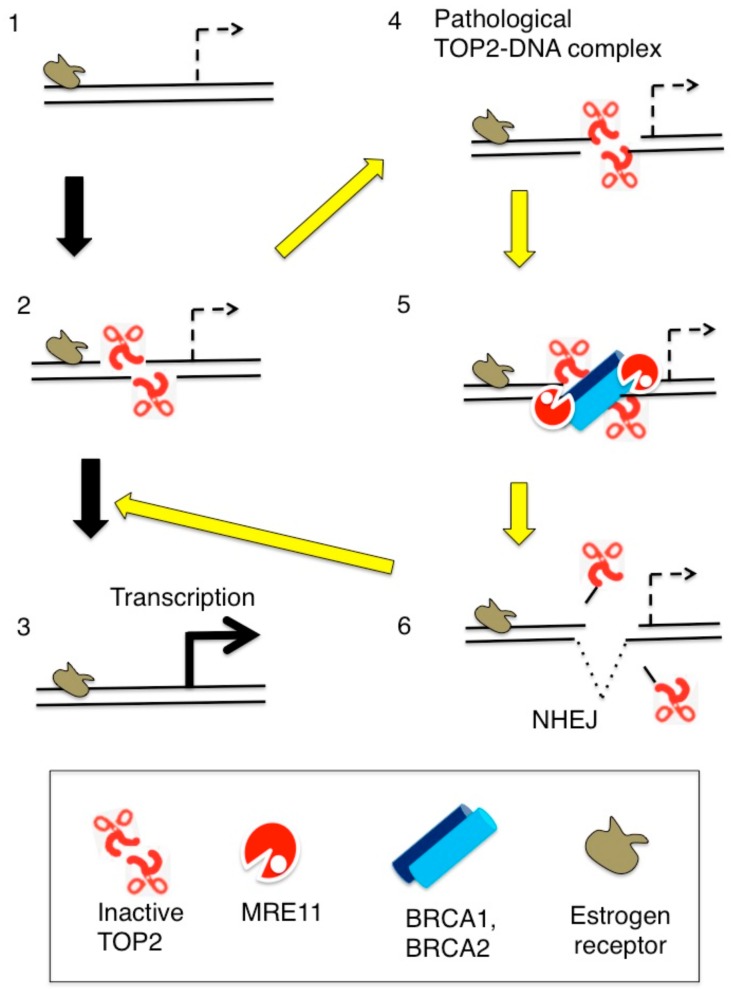
Estrogen-induced TOP2B catalysis causes the formation of stalled TOP2Bccs presumably at transcriptional regulatory sequences. (Step 1) Estrogen receptors (ER) bind at their target genes. (Step 2) TOP2B catalysis at transcriptional regulatory sequences, and (Step 3) estrogens upregulate the transcription of ER-target genes. (Step 4) TOP2B undergoes abortive catalysis generating stalled TOP2Bccs. (Step 5) From the resulting lethal DSBs TOP2B adducts are removed by the coordinated action of BRCA1, BRCA2, and Mre11. (Step 6) NHEJ seals DSBs leading a cycle of TOP2B-mediated catalyses (Step 2).

**Figure 7 genes-10-00868-f007:**
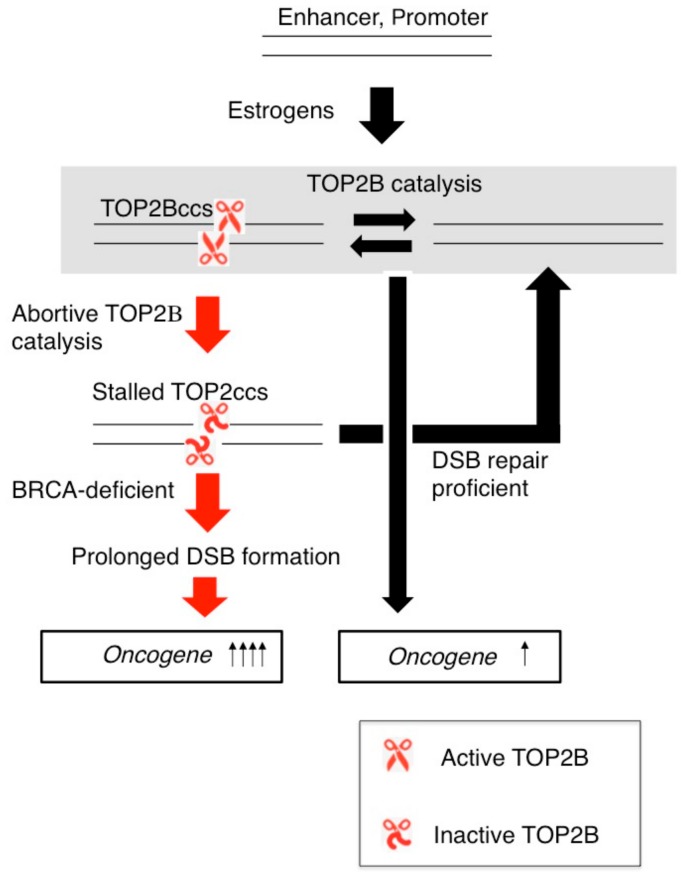
TOP2B catalysis and the formation of TOP2ccs affect transcriptional responses to estrogens. TOP2B catalysis changes the three-dimensional structure of ER-target genes such as *C-MYC* and induces their transcription. DSB-repair-proficient normal cells are capable of very quickly repairing stalled TOP2Bccs. The loss of BRCA1 or BRCA2 protein causes accumulation of unrepaired DSBs, which may upregulate the transcriptional response of oncogenes to estrogens.

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
