# Peer review of "Type II DNA Topoisomerases Cause Spontaneous Double-Strand Breaks in Genomic DNA"

_genes, 2019, doi:10.3390/genes10110868_

Round 1

Reviewer 1 Report

Comments to authors

The article by Morimoto et al reviews the role of type II DNA topoisomerase enzymes (TOP2) in physiological conditions and in the formation of pathological complexes (TOP2ccs). The article is of interest because it reviews the most important aspects of this topic based on the most relevant and current bibliography in this regard. However, I have minor comments that must be taken into consideration to improve the quality of the article and then I would recommend it to be published in Genes journal.

Line 21: the abbreviation DBS appears before it is defined in the following sentence

Line 46: there is a “strange” symbol after Step

Lines 78, 79, 80, 190, 191, 192, 193, 195, 200, 202, 209, 212, 217, 219, 237, 238, 243, 244, 248, 253, 254, 257, 318, 325, 328, 330, 331, 332, 333, 342, 359, 360, 362, 365, 391 and 395 there is a “strange” symbol after TOP

Line 317 there is a “strange” symbol after ER, I guess it could be “alpha and beta”

Reviewer 2 Report

Morimoto et al: Type II DNA topoisomerase enzymes cause spontaneously arising double-strand breaks in the genomic DNA

Originality/Novelty: This review provides a good summary of the current state of the field. Significance: The topics and studies highlighted in this review are of high significance, and therefore this review is timely. Quality of Presentation: Presentation quality is good. Figures are generally satisfactory, but the clarity and accuracy of the text needs to be improved in multiple places. Scientific Soundness: The studies and data be referenced in this review are scientifically sound. Interest to the Readers: This review will be of interest to readers from multiple disciplines, including genetics, biochemistry and cancer biology. Overall Merit: There is overall benefit to publishing this work, as it highlights recent results and outstanding questions that span multiple disciplines. English Level: There are many areas of the manuscript that require work, specifically corrections to sentence structure and punctuation are required. I have attempted to make major corrections (especially to section titles) but there are additional sentences in the manuscript that need to be rewritten in order to improve readability. Additionally, there are several places where the statement made is not quite accurate and needs to be rewritten – these are highlighted in the ‘specific comments’ section.

Brief summary:

Morimoto et al. provide a summary of our current understanding of how the action of Topoisomerases (both I and II), in relieving torsional stress in DNA, can lead to the formation of spontaneous DNA double strand breaks. They discuss, in detail, the multiple repair mechanisms present in the cell that act to process this DNA damage and restore genome stability. This discussion is extended to a detailed description of how the sex hormones can induce TOP2 catalysis and the formation of DSBs during transcription.

Broad comments:

This review by Morimoto et al. does a nice job of summarizing the important issues related to Topoisomerase action and outcomes. The amount of information included and the level of detail seems appropriate. Reference citations are satisfactory. The major weakness of this review is that the writing needs to be improved/clarified. Example and suggested changes are listed in the ‘specific comments’ section.

Specific comments:

Review Title – could be improved to make it more concise: Type II DNA topoisomerases cause spontaneous double strand breaks in genomic DNA

Abstract text lines 20-21: Change ‘Thus, this ‘abortive catalysis’ is a major cause…’ to Thus, this abortive catalysis can be a major source…

Page 1 Introduction, lines 37-38: Use of the terms ‘multiple lethal’ and ‘in every cell cycle’ don’t make sense in the following sentence: They include damaged nucleotides on template strands and single-strand breaks. Homology- directed repair (HDR) rejoins multiple lethal DSBs using the homologous sister chromatid as the 37 template DNA (Figure 1) in every cell cycle”. It could be changed to … homology-directed repair (HDR) heals DSBs using the homologous sister chromatid as the template DNA for repair.

Page 1 line 41: Change ‘The major question…’ to: One major question…

Page 2 section 2 lines 50-51: In order to better reflect the order of discussion in the text, the title could be changed to: The consequences of failure of catalysis by DNA topoisomerases I (TOP1) and II (TOP2)

The Greek symbols for TOP2a and TOP2b are not visible in the PDF text. Page 3 line 76: The sentence ‘Like TOP1, TOP2 seems to be frequently trapped on genomic DNA…’ is unclear. It could be changed to: Like TOP1, TOP2 can be trapped on genomic DNA…

Page 3 lines 93-94: Sentence should be changed to: “Nonetheless, it was previously believed that the abortive catalysis of TOP2 occurs extremely rarely under physiological conditions.”

Page 4 line 101: The following sentence is confusing: ‘… and the crucial role played by Mre11 in removing these lethal DSBs.’ It could be changed to: … the crucial role that Mre11 plays in repair of DSBs.

Page 4 Line 104 section 3 title is unclear. It could be changed to: Removal of 5’ TOP2 adducts to restore ligatable DNA ends is necessary for DSB repair by NHEJ

Figure 3 – the shapes representing proteins are too small to see clearly. It would be helpful for the authors to include a key that indicates the protein and the shape/color that it is referring to (like they do in Figure 6). Also, what does the dotted line in part C represent?

Page 4 line 116 - Define NHEJ upon first use in the text.

Page 4 Lines 118-120: The sentence beginning with ‘The choice between HDR or NHEJ..’ has several problems. It can be changed to: ‘The choice between DSB repair pathway HDR versus NHEJ can be mediated by resection, to produce 3’ single strand (ss) tails at DSBs. This resection promotes HDR and suppresses NHEJ.

Page 5 lines 131-132: In the discussion of the final step of NHEJ mediated by DNA ligase 4, it is important to cite recent work by Pryor et al. (2018: PMID 30213916) where they demonstrate that Lig4 activity is promoted by the presence of a ribonucleotide at the DSB. Therefore, although there is still a 5’ phosphate and a 3’ hydroxyl reside present, the presence of a reactive 2’OH on the ribose moiety means that the DNA ends are not the conventional ones that we think of as ‘clean’.

Page 5 line 134. The statement that “The removal of Tdp2 requires preceding digestion of 5’TOP2 adducts by the proteasome” not entirely accurate. Schellenberg et al (ref 33) demonstrated that the ZATT SUMO ligase binds to TOP2cc to facilitate a proteasome-independent Tdp2 hydrolase activity. Therefore, although proteosomal degradation can preceed Tdp2 activity on a TOP2-generated adduct, direct resolution of TOP2cc can occur without the need for proteosomal degradation, and this should also be discussed by the authors in this review.

Page 6 lines 181-182. The statement that: ‘These data indicate that the abortive catalysis of TOP2 may occur so frequently that stabilized TOP2ccs generate lethal DSBs in virtually every cell cycle’ is an overstatement. This could be changed to: …stabilized TOP2ccs have the potential to lead to the generation of lethal DSBs.

Page 8 line 249. The sentence: ‘Thus, the validity of the first hypothesis remains proved’ is incorrect. Please change it to: Thus, the validity of the first hypothesis remains to be proven.

Page 8 line 251. The sentence ‘…loss of DNA-PKcs caused a few times…’ should be changed by removing ‘a few times’.

Page 9 line 280. Describe how an R-loop is formed when you first begin referring to these structures in the text.

Page 9 line 286: Indicate here that Ribonuclease H selectively digests the RNA portion of an R-loop.

Page 9 line 299: The sentence ‘… BRCA1 quickly accumulates at etoposide-induced DNA damage sites even in the G1 phase through HDR is suppressed in the G1 phase’ is very unclear and needs to be rewritten. I believe that the authors meant that despite the fact that HDR is suppressed, BRCA1 still accumulates at etoposide-induced DNA damage sites.

Page 11 line 352: Change ‘…whole cell cycle phase’ to ‘throughout the cell cycle…’

Page 11 line 354: Pre-subscribed should be ‘prescribed to treat’.

Page 11 line 356: Change ‘…instantaneous removal…’ to ‘efficient repair of DSBs’.

Page 12 line 388-389: Change ‘…insertion and deletion at joint sequences’ to ‘insertion and deletion mutations forming at joint/junction sequences’.

Page 12 lines 390-392: The sentence ‘… the widely accepted hypothesis that TOP2 functions in resolving positive supercoiling ahead of elongating or paused PolII needs to be validated in vivo’ should be removed as this function of TOP2 is not discussed in detail in this review.

Reviewer 3 Report

This is a informative review of the roles of TOP2 in in DSB formation, with a focus on the transcription connection.  While BRCA proteins have a well characterized role in promoting HR, a potential role in NHEJ as well is intriguing.  This will be a valuable addition to the field.
